# Predicting the Development of Anti-Drug Antibodies against Recombinant alpha-Galactosidase A in Male Patients with Classical Fabry Disease

**DOI:** 10.3390/ijms21165784

**Published:** 2020-08-12

**Authors:** Sanne J. van der Veen, Wytze J. Vlietstra, Laura van Dussen, André B.P. van Kuilenburg, Marcel G. W. Dijkgraaf, Malte Lenders, Eva Brand, Christoph Wanner, Derralynn Hughes, Perry M. Elliott, Carla E. M. Hollak, Mirjam Langeveld

**Affiliations:** 1Department of Endocrinology and Metabolism, Amsterdam University Medical Centers (AUMC), University of Amsterdam, 1105 AZ Amsterdam, The Netherlands; l.vandussen@amsterdamumc.nl (L.v.D.); c.e.hollak@amsterdamumc.nl (C.E.M.H.); m.langeveld@amsterdamumc.nl (M.L.); 2Department of Medical Informatics, Erasmus University, 3000 CA Rotterdam, The Netherlands; wytze.vlietstra@gmail.com; 3Laboratory Genetic Metabolic Diseases, Amsterdam University Medical Centers (AUMC), University of Amsterdam, 1105 AZ Amsterdam, The Netherlands; a.b.vankuilenburg@amsterdamumc.nl; 4Department of Epidemiology and Data Science, Amsterdam University Medical Centers, University of Amsterdam, 1105 AZ Amsterdam, The Netherlands; m.g.dijkgraaf@amsterdamumc.nl; 5Department of Internal Medicine D, and Interdisciplinary Fabry Center (IFAZ), University Hospital Muenster, D-48149 Muenster, Germany; malte.lenders@ukmuenster.de (M.L.); eva.brand@ukmuenster.de (E.B.); 6Division of Nephrology, Department of Medicine, University Hospital Würzburg, 97080 Würzburg, Germany; wanner_c@ukw.de; 7Department of Haematology, Royal Free London National Health Service Foundation Trust and University College London, London NW3 2QG, UK; derralynnhughes@nhs.net; 8Department of Cardiology, St. Bartholomew’s Hospital and University College London, London EC1A 7BE, UK; perry.elliott@ucl.ac.uk

**Keywords:** Fabry disease, enzyme replacement therapy, anti-drug antibodies, prediction model

## Abstract

Fabry Disease (FD) is a rare, X-linked, lysosomal storage disease that mainly causes renal, cardiac and cerebral complications. Enzyme replacement therapy (ERT) with recombinant alpha-galactosidase A is available, but approximately 50% of male patients with classical FD develop inhibiting anti-drug antibodies (iADAs) that lead to reduced biochemical responses and an accelerated loss of renal function. Once immunization has occurred, iADAs tend to persist and tolerization is hard to achieve. Here we developed a pre-treatment prediction model for iADA development in FD using existing data from 120 classical male FD patients from three European centers, treated with ERT. We found that nonsense and frameshift mutations in the α-galactosidase A gene (*p* = 0.05), higher plasma lysoGb3 at baseline (*p* < 0.001) and agalsidase beta as first treatment (*p* = 0.006) were significantly associated with iADA development. Prediction performance of a Random Forest model, using multiple variables (AUC-ROC: 0.77) was compared to a logistic regression (LR) model using the three significantly associated variables (AUC-ROC: 0.77). The LR model can be used to determine iADA risk in individual FD patients prior to treatment initiation. This helps to determine in which patients adjusted treatment and/or immunomodulatory regimes may be considered to minimize iADA development risk.

## 1. Introduction

Fabry disease (FD; OMIM 301500) is a rare, X-linked, lysosomal storage disease caused by mutations in the alpha-galactosidase A (*GLA*) gene. This leads to absent or reduced alpha-galactosidase A enzyme activity and the subsequent accumulation of its substrate globotriaosylceramide (Gb3). Accumulation of Gb3 and its deacylated form globotriaosylsphingosine (lysoGb3) results in progressive damage to heart, kidneys and brain [1]. The disease is most often treated with biweekly infusions of recombinant alpha-galactosidase A (r-αGAL A), also referred to as enzyme replacement therapy (ERT). Two ERT preparations are currently available. One is agalsidase beta (Fabrazyme, Sanofi Genzyme), produced in Chinese hamster ovary cells (CHO) and most often dosed at 1 mg/kg biweekly. The other is agalsidase-alfa (Replagal, Takeda), produced in human fibroblasts and dosed at 0.2 mg/kg biweekly. In patients with the most severe disease phenotype, male patients with classical FD, treatment with ERT often results in the development of anti-drug antibodies (ADAs). Because these patients have little to no native enzyme, the immune system recognizes the exogenously administered enzyme as foreign. ADAs are thought to be responsible for both the infusion-related reactions including fever, chills and chest pain as well as the more classical allergic reactions with edema, dyspnea, rash, itching and (rarely) anaphylactic shock [2,3,4]. In approximately half of the classical male FD patients, the formed ADAs are capable of inhibiting αGAL A activity (iADAs) in vitro [5,6] as well as inhibiting enzyme uptake into cells [7]. In vivo, iADAs negatively influence pharmacokinetics of the recombinant enzyme [5,8,9] and -titer dependently- limit the biochemical response to treatment with ERT [6]. Additionally, the development of ADAs is linked to incomplete clearance of substrate in endothelial cells or re-accumulation after initial clearance [10,11]. Clinically, male FD patients with established iADAs had higher disease severity scores [12], although the inclusion of male patients with classical as well as non-classical disease may have influenced the outcome. The negative effect of iADAs on disease outcome was confirmed in classical male patients, as iADA positive patients had an accelerated renal decline compared to iADA negative patients [6].

Experience in other disorders in which treatment with recombinant proteins is hampered by iADA formation (e.g., hemophilia, Pompe disease and MPS1) [13,14,15,16], shows that once iADAs occur, they tend to persist despite treatment with immunosuppressants or tolerization protocols (e.g., immune tolerance is hard to achieve) [17,18,19,20]. Therefore, it is important to develop protocols that can prevent or treat iADAs. Methods to prevent iADA development should primarily be tested in patients with a high risk of developing these antibodies. Comparing different pre-treatment immunomodulatory interventions on high-risk patients, rather than all Fabry patients, is expected to improve the efficiency and interpretation of these future studies, and reduce the required sample sizes.

Several variables were found to be related to the immunogenicity of other biotherapeutics. These included: the dose and frequency of administration [21], the origin and glycosylation of the product (e.g., in which cell line is the product produced) [22,23], the age of the recipient [21,24], mutation type (nonsense vs. missense) [25] and the presence of residual native protein in the patient, also referred to as cross- reactive immunologic material (CRIM) status [15,25]. These factors may play a role in iADA formation in Fabry disease as well. Within our population of male Fabry patients with classical disease we noticed disparity in the risk of iADA formation, with families with high and low risk, suggesting genetic predilection. In this study we set out to answer the following two questions: (1) Which factors predispose male patients with classical FD for iADA development? (2). How accurately can the development of iADAs be predicted prior to treatment initiation in an individual patient? To address these questions we used previously collected [23] demographic, medical and biochemical data from 120 classical male FD patients and used these data to build and validate predictive models.

## 2. Results

### 2.1. Patient Characteristics

Patient characteristics of the 120 included patients are outlined in Table 1.

### 2.2. Logistic Regression Model

To identify factors associated with iADAs and assess their ability to predict iADA formation, we built a logistic regression (LR) model, using backwards selection of variables. The final model was validated internally using repeated cross validation. In this model, three variables were included that were found to be associated with an increased risk for iADA development: higher levels of the biomarker lysoGb3 before start of treatment (*p* < 0.001), the presence of a nonsense or frameshift mutation (*p* = 0.053) and starting treatment with agalsidase beta (*p* = 0.006). Distribution of individual variables is visualized in Figure 1a–c. Predictive performance was assessed with the area under the receiver operating characteristic curve (AUC-ROC) which was 0.77 (Figure 2a). Optimal accuracy (0.73, Figure 2b) was determined at a cutoff of 0.53 (sensitivity: 0.69, specificity: 0.76, Figure 2c,d). Figure 2e shows the predicted vs. observed outcome for each patient.

Age at start treatment did not influence the risk of iADA development. Both initial treatment type (agalsidase alfa vs. agalsidase beta) and treatment dose (0.2, 0.5 or 1.0 mg/kg every other week) were significantly associated with a higher risk for iADA development. Since these variables are strongly related, only treatment type was included in the model. Location of the mutation as a numeric factor was not associated with an increased risk. However one location of missense mutations seemed especially prone for iADA development and is explored separately in a post-hoc analysis.

### 2.3. Random Forest Model

A second model was built using the ensemble learning method ‘Random Forest’ (RF). Ensemble learning models can deal with co-linearity and are able to handle many variables. Furthermore, these models can handle variables with relative low contribution to the calculated risk. We build a RF model using the following variables: baseline plasma lysoGb3, mutation type and location, age of start, first treatment type and first treatment dose. Compared to the LR model, the fit of the RF model was similar (AUC-ROC 0.77) and it did not improve prediction accuracy. For details see Appendix A.

### 2.4. Post Hoc Analyses

Patients with missense mutations were less likely to develop iADAs compared to patients with nonsense and frameshift mutations. To check whether, within the missense group, the location of the mutation influenced iADA risk, we visualized the location of missense mutation and the presence of iADAs (Appendix A). Mutation location as a numeric variable was not associated with iADA risk. However, eight out of 11 patients with a mutation at the c.1025 position developed iADAs (73%) compared to 17 out of 59 (29%) of patients with missense mutations at other positions (*p* = 0.01, OR 6.3). Most other mutations in this cohort were unique mutations, thus an effect of location for these mutations could not be established. In silico analysis did not suggest that variants at position c.1025 affected splicing of the pre-mRNA *GLA*.

### 2.5. Second Cohort

Due to differences in the iADA detection method (sample dilution vs. enzyme saturation) as well as lysoGb3 analyses (dried blood spot vs. plasma), we were not able to test our model on the second, independent, cohort of patients. We therefore performed the same steps for imputation and build a LR prediction model using the same three variables on this second cohort to check reproducibility of the results. In this group of 30 classical male patients, only mutation-type was significantly correlated (*p* = 0.01) to the risk of iADA development. Initial treatment type (*p* = 0.3) and baseline lysoGb3 (*p* = 0.4) were not significantly related to iADA risk in this smaller cohort. For details see Appendix A.

## 3. Discussion

Our large international Fabry cohort study identified the following variables to be associated with an increased risk of iADA development in male patients with classical FD: (1) having a nonsense or frameshift mutation. (2) starting treatment with agalsidase beta and (3) higher levels of the disease biomarker (lysoGb3) in plasma before start of treatment. In the second control cohort, the importance of mutation type was confirmed, but type of recombinant enzyme and plasma lysoGb3 levels showed only a trend, which may have been due to the smaller sample size and/or the differences in lysoGb3 measurements.

Mutation type has previously been found to be related to iADA risk in FD [6,26], which is in accordance with observations in Pompe disease and hemophilia [27,28]. Although all classical patients have minimal or no residual enzyme activity, it is possible that the mutated protein is still produced in small amounts. Thus, the immune system of patients with missense mutations may still be exposed to the protein, leading to central tolerance induction. In patients with a large deletion or early frameshift mutations, the protein will either not be produced at all, or it will be truncated. These patients are less likely to develop central tolerance and are more likely to develop an immunological reaction to exogenously administered enzyme. The increased occurrence of iADAs in patients who started treatment with agalsidase beta vs. agalsidase alfa could either be attributed to the higher dose (1 mg/kg vs. 0.2 mg/kg), the difference in production cell-line and thus in glycosylation pattern (Chinese hamster ovary cells for agalsidase beta vs. human fibroblasts for agalsidase alfa) or a combination of both. The group of patients that started with a lower than recommended dose of agalsidase beta (see Table 1) was too small to draw conclusions on the dose effect. In this study, the plasma lysoGb3 level was identified as an independent predictive variable for iADA formation. We hypothesize that either plasma lysoGb3 levels reflect subtle differences in enzyme activity (not detectable in the enzyme activity assay) or that the pro-inflammatory effects of (lyso)Gb3 [29] serve as an adjuvant to prime the immune system.

Detrimental effects of iADAs on treatment effectiveness in FD is becoming increasingly clear over the last decade [5,7,10,11,12]. Overcoming iADA development is therefore essential to improve treatment outcome and can hypothetically be achieved in two ways: (1) by achieving tolerization through immune tolerance induction (ITI) in iADA positive patients or (2) by preventing iADA formation prior to treatment. Both strategies have previously been tried in other diseases.

The first approach, ITI, has been studied extensively in hemophilia. Most ITI protocols require long term, frequent (>3 times a week) administration of the recombinant protein and are still only successful in 60–70% of cases for hemophilia A and only 30% of cases for hemophilia B [19,30]. In Pompe disease, ITI has been tried with intensive immune-modulatory protocols (including rituximab, methotrexate, bortezomib and intraveneus immunoglobulines) resulting in a steady decrease in iADAs and improvement of therapeutic effectiveness, but full tolerization was not achieved [17,18]. This is in accordance with the findings from Lenders et al. in FD patients who underwent kidney or heart transplantation (and were thus treated with immunosuppressive therapy). Patients with established iADAs demonstrated an initial reduction in their iADA titer [31]. However, after tapering of the immunosuppressive medication (specifically corticosteroids), the iADA titers increased again [31]. Therefore, patients would require continuous exposure to immunosuppressive drugs to maintain immune tolerance, with unacceptable side effects for such a slowly progressive disease as FD.

The second approach (preventing iADA formation) has been tried using immunosuppressive medication in infantile Pompe disease and MPS1 patients simultaneously with—or shortly before— ERT initiation. This approach has proven to be difficult, as iADAs often still arise after tapering of the immunosuppressive medication [20,32,33]. This may be explained by the fact that immune suppression was not optimal at the time ERT was started (in infantile Pompe disease ERT initiation cannot be delayed). In six classical male FD patients that started ERT after a transplantation, iADAs did not develop [31].

Another approach to iADA prevention is treatment initiation with lower (and more regular) doses of recombinant protein. In hemophilia, starting treatment with lower-dosed prophylactic treatment at regular intervals was associated with a 60% lower risk of iADA development compared to patients that started treatment with high doses and continued to get high doses ‘on demand’ (e.g., at irregular intervals) [34]. This is in accordance with the findings in our study that starting treatment with the lower dosed agalsidase alfa is associated with a lower risk for iADA development. Thus starting treatment with lower than registered doses of recombinant enzyme, in combination with shortening administration intervals, might be a way to induce central tolerance. Once central tolerance is induced, doses could gradually be increased and dosing intervals reduced. Future studies will focus on finding an optimal build up schedule for this patient group.

Our prediction model was built on a selective patient group (male patients with classical FD). We chose this approach as female patients and patients with non-classical disease do not tend to develop iADAs. Using this model we are able to correctly predict iADA formation in 73% of male FD patients with the classical disease phenotype. To further optimize predictive accuracy in future models other variables could be included, that were not present in our current dataset. Studies in Pompe disease and hemophilia describe a potential influence of certain gene polymorphisms, such as the HLA haplotype [35,36,37]. In hemophilia, the presence of so-called danger signals before or during the first infusions (e.g., recent surgery, bleeds, vaccinations and active infections) were associated with an increased risk for iADA development [38]. The explanation is that the danger or stress signals that are released work as an adjuvants to induce immunogenicity. Although active infection is already a contra-indication for ERT administration, it may be wise to avoid initiating ERT soon after other stressors as well (e.g., surgery or vaccinations).

The outcome of this model can be translated to clinical care as the individual risk of iADAs in new patients helps physicians to decide which patients are eligible for pre-ERT immunomodulatory interventions. In addition, knowing the a-priori risk aides the evaluation of the effectiveness of these interventions, as the pre-intervention risk can be included in the outcome analysis.

## 4. Materials and Methods

### 4.1. Patients

This study was conducted in accordance with the principles of the Helsinki Declaration, as revised in 2013. To build the models, retrospective collected data from three European FD centers of excellence (Amsterdam University Medical Center, The Netherlands; Royal Free London NHS Foundation Trust, United Kingdom; and the University Hospital Würzburg, Germany) were used [39]. Data included basic diagnostic data, clinical and biochemical parameters, comorbidities and medication use. In this study, only male patients with a classical disease phenotype were included. Male patients were classified as having a classical phenotype based on both a residual enzymatic activity of less than 5% and the presence of one or more of the characteristic classic FD symptoms (acroparesthesia, clustered angiokeratoma, cornea verticillata), as described by Arends et al. [40]. All included patients were treated with ERT (agalsidase alfa or -beta). Patients who switched dose or treatment type were not excluded. Agalsidase alfa was always dosed at 0.2 mg/kg/eow, Agalsidase beta was predominantly dosed at 1 mg/kg/eow, but patients on 0.2 mg/kg (*n* = 4) and 0.5 mg/kg (*n* = 6) were also included.

### 4.2. Variables and Development of the Prediction Models

The Tripod checklist for prediction model development was followed where possible [41]. The following variables were included and tested: age at start ERT, mutation location and type, plasma lysoGb3 at baseline and the initial dose and type of ERT.

### 4.3. Laboratory Measurements

Plasma lysoGb3 and iADAs were measured at the AMC. Plasma lysoGb3 values were obtained within one year before start of treatment with ERT. LysoGb3 levels were analyzed using tandem mass spectrometry, as described previously [42,43]. AMC samples from before August 2015 used a different internal standard than samples from later time points at the AMC as well as samples from the Royal Free Hospital and the University Clinic Würzburg. After application of a correction factor, outcomes using both internal standards correlated well [40].

IADAs were measured as previously described [5]. In short, patient plasma in various dilutions is added to the recombinant protein. The titer represents the amount of dilutions needed to recover at least 50% of enzymatic activity in vitro. Outcome was determined as iADA positive (iADA+) or iADA negative (iADA−). Patients were considered iADA+ if they tested positive at one or more time points for αGAL A inhibition with a titer of 6 or higher. iADA titers were measured after at least 1 year of treatment, with the exception of two patients for whom only iADA titers at 9 months after ERT initiation were available. 23/120 patients had only one iADA measurement, 24/120 had two measurements, all other patients had three or more iADA measurements (range 3–16).

### 4.4. Statistics

For statistical analysis and model building, R (version 3.4.3) was used. A markdown file containing the full code and results was added as Appendix A. Package ‘MICE’ was used for data imputation. Package ‘caret’ was used to build and validate predictive models. Package ‘ggplot2’ was used for data visualization.

#### 4.4.1. Algorithms

The performance of two different machine learning algorithms for the prediction of iADAs were compared: Logistic Regression (LR) and the most common ensemble method, i.e., Random Forest (RF). Overall LR and RF are known to give comparable results, but accuracy may vary depending on the number of subjects, number of explanatory variables and amount of noise variables [44]. LR is a form of binomial regression and models the probabilities for classification problems with two possible outcomes, in this case iADA+ and iADA−. It uses the logarithm of the odds (i.e., the logarithm of the probability of iADA+ divided by the probability of iADA− status), resulting in a linear combination of the independent variables (predictors) (Figure 3a). All assumptions for LR were met, details can be found in the Appendix A.

RF utilizes “ensemble learning”. In brief, it creates many decision trees from random samples and averages out the results to get a clear model (Figure 3b). Unlike LR, RF does not assume a linear relationship between variables and therefore outcome and model are not influenced by co-linearity. Furthermore RF models can handle many different variables and do not require a significant contribution to the predicted risk for individual variables [44].

#### 4.4.2. Handling of Missing Data

Missing data (3%) consisted mostly of missing lysoGb3 values (28% of lysoGb3 values were missing) and were imputed using multiple imputation (package MICE) [45]. It is important to note that the MICE algorithm assumes missing data to be either Missing At Random (MAR), meaning that the probability that a value is missing can be explained by the observed values, or Completely Missing At Random (CMAR), meaning that there is no other reason for missing data than chance. In our study the treatment center and date of start ERT were the main explanatory variables for missing lysoGb3 measurements. We assumed that the year of ERT initiations and the location where patients were treated did not influence the risk for iADA development and thus considered the data to be MAR. To assess reproducibility, imputation was repeated five times and distribution of imputed variables as well as differences in mean and SD for the imputed variables were visualized and added in the Appendix A. Five individual imputed datasets were created, each consisting of 5 cycles. Imputation was deemed consistent based on the comparable means and low standard deviation for the imputed values. Imputed datasets were merged and the mean of all 5 imputations was used as the final imputed value.

#### 4.4.3. Experiment and Intrinsic Validation

For a detailed description of steps and full code see the Appendix A. In short R package CARET was used to build two prediction models. To prevent overfitting, without losing data, validation was done using 10-fold cross validation, meaning that the dataset was divided into 10 subsets. For every round nine folds were used to train the model and this model was then validated on the remaining folds. This was repeated until every set was used in both the training as the testing group. These steps were then repeated 10 times and results of each individual patient were aggregated to form the final model (Figure 3c). All variables used in the LR model met LR assumptions (visualization in SM.1). The area under the curve (AUC) of the receiver operating characteristic curve (ROC) was used to compare classification performance. The cutoff was decided based on optimal accuracy.

### 4.5. External Cohort

A second dataset of 30 classical male patients was provided by the University Hospital Muenster, Germany to validate the results of our model. Phenotyping was done locally and based on classical FD symptoms and enzyme activity. In six patients, no baseline enzyme activity was available, all presented with classical symptoms and had a mutation associated with classical disease. Patients were treated with agalsidase alfa (dosed at 0.2 mg/kg/eow) or agalsidase beta (dosed at 0.2–1.37 mg/kg/eow). LysoGb3 measurements were performed at Centogene (Rockstock, Germany) in dry blood-spot. Lyso-Ceramide was used as reference (Matreya, LLC, Pleasant Gap, PA, USA) and D5-Fluticasone Propionate (EJY Tech, Inc., Rockville, MD, USA) served as internal standard. LysoGb3 values at baseline were missing in 10/30 patients. IADAs were measured as described previously [29]. All patients with >50% inhibition are considered iADA positive. Titers represent the amount of enzyme necessary to overcome the neutralizing capacity of iADAs in a patient. Due to differences in measuring techniques for both iADA status, lysoGb3 and differences in ERT dosing the data could not be used to validate the initial model, to show reproducibility model building was repeated with the same variables and outcome is reported separately.

## Figures and Tables

**Figure 1 ijms-21-05784-f001:**
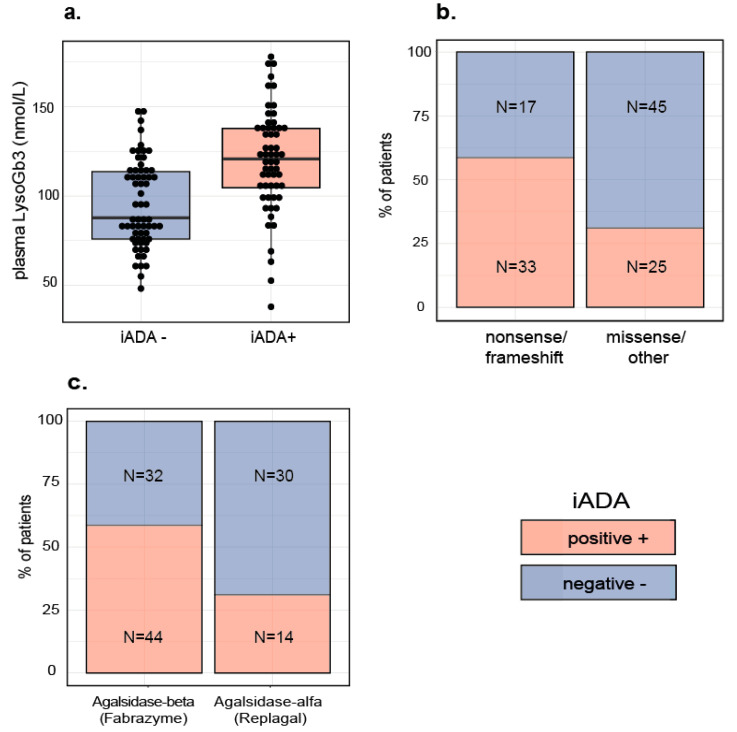
Distribution of variables significantly associated with an increased risk of iADA development in male Fabry patients with classical disease. Color represents iADA status. (**a**) Baseline plasma lysoGb3 levels. (**b**) Mutation type (n.b. eight out of 25 iADA positive patients in the missense group had a mutation at location c.1025). (**c**) Treatment type at start of treatment: agalsidase beta (0.2–1 mg/kg) versus agalsidase alfa (0.2 mg/kg).

**Figure 2 ijms-21-05784-f002:**
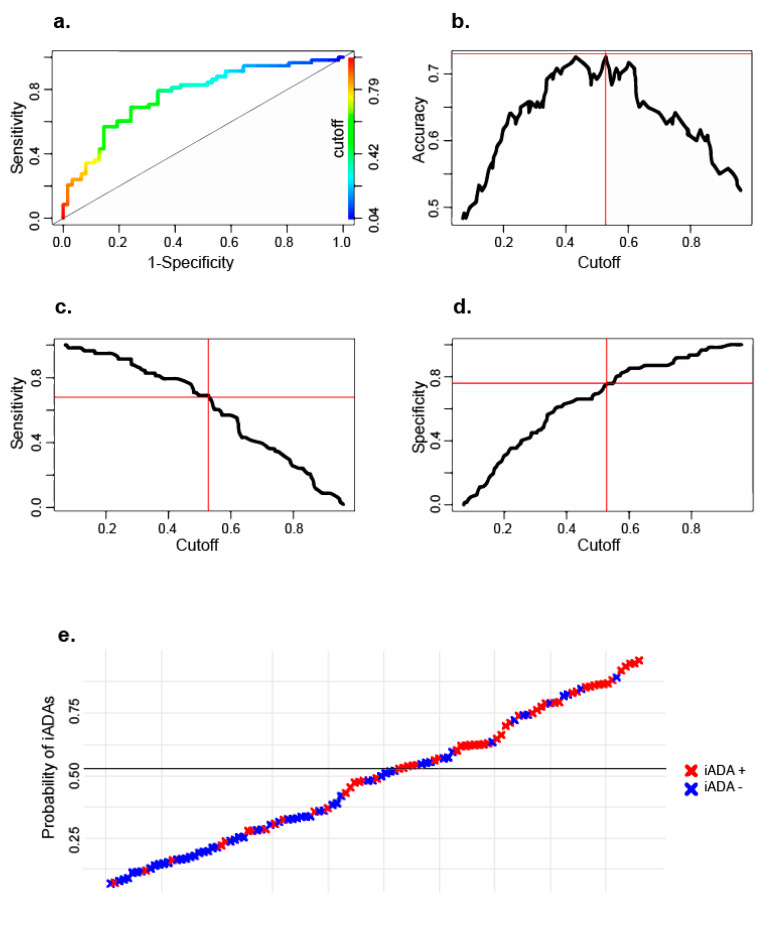
Results from the LR model. (**a**) ROC curve, AUC = 0.77. Colors represent the different possible cutoff* values; (**b**) Accuracy of the model at different cutoffs* (maximum of 0.73 at cutoff 0.53); (**c**) Sensitivity of the model. At the chosen cutoff sensitivity is 0.69; (**d**) Specificity of the model at the chosen cutoff is 0.76; (**e**) Visualization of the predicted (Y axis) versus observed outcome (color coded) per patient. The line drawn shows the chosen cutoff. * The cutoff is a chosen decision threshold above which patients are predicted as positive (will develop ADAs). Lower cutoffs favor sensitivity, higher cutoffs favor specificity.

**Figure 3 ijms-21-05784-f003:**
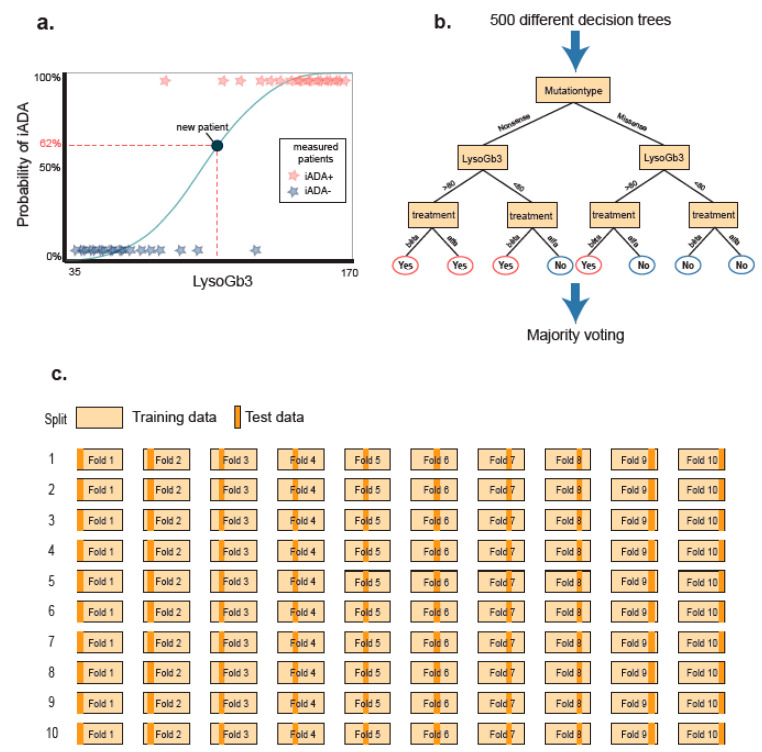
Visualization of used methods. (**a**) Logistic regression uses a combination of independent variables to draw a sigmoid curve that fits best with the training data. New subjects are plotted on the curve to calculate the risk of iADA development (in schematic presentation this is based on a single variable (lysoGb3), in reality all contributing variables weigh in in the predicted outcome). Blue stars resemble iADA negative subjects and the red stars represent iADA positive patients in the data set used to build the model. The blue dot represents a subject in the test set. (**b**) Random forest is a classification algorithm. It randomly creates multiple decision trees (default is 500). Each tree results in a conclusion (e.g., iADA yes or no), majority voting of all trees is used to determine risk of iADA development; (**c**) Cross-validation is a resampling procedure used to evaluate predictive models with limited data. The goal is to optimize usage of data and minimize overestimation of predictive accuracy. The data was randomly split in 10 subsets. For each iteration 9 sets are used to build the model and one to test the model, until every subject has been in both groups. This procedure is repeated 10 times, until 100 models are built. Outcome of each individual patient were averaged and used to build the final model.

**Table 1 ijms-21-05784-t001:** Characteristics of 120 male patients with classic Fabry disease.

	iADA+	iADA−
**Site (N, % of total)**		
- Amsterdam UMC	23 (40%)	16 (26%)
- The Royal Free Hospital	24 (41%)	26 (42%)
- Universitätsklinikum Würzburg	11 (19%)	20 (32%)
**Mutation type (N, % of total)**		
- Nonsense/frameshift	33 (57%)	17 (27%)
- Missense	21 (36%)	37 (60%)
- Other	4 (7%)	8 (13%)
**Age at ERT start (years, median, range)**	37 (9–58)	35 (13–63)
**LysoGb3 (nmol/L, median, range)**	123 (38–178)	96 (48–149)
**First treatment (N, % of total)**		
- Agalsidase alfa 0.2 mg/kg	14 (24%)	31 (50%)
- Agalsidase beta 0.2 mg/kg	4 (7%)	2 (3%)
- Agalsidase beta 0.5 mg/kg	2 (3%)	2 (3%)
- Agalsidase beta 1.0 mg/kg	38 (66%)	27 (44%)
**Inhibition titer (median, range)**	113 (7–32645)	0 (0–5)

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
