# Peer review of "Predicting the Development of Anti-Drug Antibodies against Recombinant alpha-Galactosidase A in Male Patients with Classical Fabry Disease"

_ijms, 2020, doi:10.3390/ijms21165784_

Round 1
Reviewer 1 Report
In this study the authors developed an algorithm to identify among the patients with Fabry Disease those who have the highest probability of developing anti-drug antibodies when are treated with ERT. This study is well done and it will be important for patient management in the future. In my opinion can be published in the International Journal of Molecular Sciences in is present format.
However:
- One of the approaches proposed in the manuscript to reduce the probability of developing iADA is to reduce ERT doses in the high-risk patients and at the same increase its frequency of administration. Nevertheless, the authors should be clearer, specifying which doses are considering (ranges for variation for example) as well as the frequency of administration. This for each ERT preparation already in the market. The authors should also mention that these doses/frequencies of administration are not the approved doses/frequencies for these ERT preparations.
Minor point:
The authors report a strong association between missense mutations at position c.1025 and iADA. Please verify if mutations in this position could also affect splicing of the pre-mRNA GLA.
Author Response
Dear reviewer,
Thank you for your kind words.
Comment 1:
One of the approaches proposed in the manuscript to reduce the probability of developing iADA is to reduce ERT doses in the high-risk patients and at the same increase its frequency of administration. Nevertheless, the authors should be clearer, specifying which doses are considering (ranges for variation for example) as well as the frequency of administration. This for each ERT preparation already in the market. The authors should also mention that these doses/frequencies of administration are not the approved doses/frequencies for these ERT preparations.
Response to comment 1
We are in the process of setting up a study to identify the optimal strategy based on (the limited) experience in other diseases and are not yet ready to elaborate in more detail about a potential protocol. We did however change the following in our manuscript:
At page 7 (line 213) we changed:
“Thus starting treatment with lower doses of recombinant enzyme, possibly at shorter intervals, might be a way to induce central tolerance. Once central tolerance is induced, doses could be increased.”
To
“Thus starting treatment with lower than registered doses of recombinant enzyme, in combination with shortening administration intervals, might be a way to induce central tolerance. Once central tolerance is induced, doses could gradually be increased and dosing intervals reduced. Future studies will focus on finding an optimal build up schedule for this patient group.”
Comment 2:
The authors report a strong association between missense mutations at position c.1025 and iADA. Please verify if mutations in this position could also affect splicing of the pre-mRNA GLA.
Response to comment 2
We checked in silico analyses on the mutations at position c.1025 and found no evidence that suggests that the splicing is affected.
We therefor added the following sentence on page 6 (line 147):
“In silico analysis did not suggest that variants at position c.1025 affected splicing of the pre-mRNA GLA.”
Kind regards,
Sanne Jolien van der Veen
Reviewer 2 Report
Fabry disease is an X-linked deficiency of the lysosomal enzyme α-galactosidase A, which is required for normal globotriaosylceramide catabolism. Аccumulation of globotriaosylceramide and its deacylated form (lysoGb3) causes damamge to many tissues (for example, vascular endothelium, lymphatic vessels, heart, kidneys). Enzyme replacement therapy with recombinant alpha-galactosidase A is the main approach for Fabry disease treatment, but inhibiting anti-drug antibodies reduce the effectiveness of such therapy in almost 50% of patients. It was shown that mutations in the alpha-galactosidase A gene, plasma lysoGb3 level and agalsidase-beta as first treatment were significantly associated with anti-drug antibodies development. Basing on this authors developed pre-treatment prediction model for anti-drug antibodies production during treatment. I think that the study carried out at a very high level, the methods used are credible. The model proposed by the authors can be translated to clinical care thereby making a great contribution to the development of methods of therapy for this and other rare genetic diseases. This approach will significantly improve the quality of life of patients. I believe that the article should be accepted for publication in a present form.
Author Response
Dear reviewer,
We would like to thank you for your kind words and for taking the time to review our paper. We will indeed be working on translating this model into clinical care and assess tolerazation strategies for Fabry patients.
Kind regards,
Sanne Jolien van der Veen